# DHOG: DEEP HIERARCHICAL OBJECT GROUPING

## ABSTRACT

Unsupervised learning of categorical representations using data augmentations appears to be a promising approach and has proven useful for finding suitable representations for downstream tasks. However current state-of-the-art methods require preprocessing (e.g. Sobel edge detection) to work. We introduce a mutual information minimization strategy for unsupervised learning from augmentations, that prevents learning from locking on to easy to find, yet unimportant, representations at the expense of more informative ones requiring more complex processing. We demonstrate specifically that this process learns representations which capture higher mutual information between augmentations, and demonstrate that these representations are better suited to the downstream exemplar task of clustering. We obtain substantial accuracy improvements on CIFAR-10, CIFAR-100-20, and SVHN.

## 1 INTRODUCTION

It is very expensive to label a dataset with respect to a particular task. Consider the alternative where a user, instead of labelling a dataset, specifies a simple set of class-preserving transformations or 'augmentations'. For example, lighting changes will not change a dog into a cat. Is it possible to learn a model that produces a useful representation by leveraging a set of such augmentations? This representation would need to be good at capturing salient information about the data, and enable downstream tasks to be done efficiently. If the representation were a discrete labelling which groups the dataset into clusters, an obvious choice of downstream task is unsupervised clustering. Ideally the clusters should match direct labelling, without ever having been learnt on explicitly labelled data.

Using data augmentations to drive unsupervised representation learning for images has been explored by a number of authors (Dosovitskiy et al., 2014; 2015; Bachman et al., 2019; Chang et al., 2017; Wu et al., 2019; Ji et al., 2019; Cubuk et al., 2019). These approaches typically involve learning neural networks that map augmentations of the same image to similar representations, which is reasonable since variances across many common augmentations often align with the invariances we would require.

A number of earlier works target maximising mutual information (MI) between augmentations (van den Oord et al., 2018; Hjelm et al., 2019; Wu et al., 2019; Ji et al., 2019; Bachman et al., 2019). Targetting high MI between representations computed from distinct augmentations enables learning representations that capture the invariances induced by the augmentations. We are interested in a particularly parsimonious representation: a discrete labelling of the data. This labelling can be seen as a clustering (Ji et al., 2019) procedure, where MI can be computed and assessment can be done directly using the learned labelling, as opposed to via an auxiliary network trained posthoc.

### 1.1 SUBOPTIMAL MUTUAL INFORMATION MAXIMISATION

We argue and show that the MI objective is not maximised effectively in existing work due to the combination of:

1. **Greedy optimisation algorithms** used to train neural networks, such as stochastic gradient descent (SGD) that potentially target local optima; and

2. A limited set of data augmentations that can result in the existence of multiple **local optima to the MI maximisation** objective.

SGD is greedy in the sense that early-found high-gradient features can dominate and so networks will tend to learn easier-to-compute locally-optimal representations (for example, one that can be computed using fewer neural network layers) over those that depend on complex features.

By way of example, in natural images, average colour is an easy-to-compute characteristic, whereas object type is not. If the augmentation strategy preserves average colour, then a reasonable mapping need only compute colour information, and high MI between learned image representations will be obtained. This result is suboptimal in the sense that a hypothetical higher MI optima exists that also captures semantic information, assuming the model has sufficient capacity to learn and represent this. The conceivable existence of many such local optima coupled with greedy optimisation presents a challenge: *how can we leverage powerful image augmentation-driven MI objectives while avoiding greedily-found local optima*?

**Dealing with greedy solutions**   Heuristic solutions, such as as Sobel edge-detection (Caron et al., 2018; Ji et al., 2019) as a pre-processing step, have been suggested to remove/alter the features in images that may cause trivial representations to be learned. This is a symptomatic treatment and not a solution. In the work presented herein, we acknowledge that greedy SGD can get stuck in local optima of the MI maximisation objective because of limited data augmentations. Instead of trying to prevent a greedy solution, our technique lets a model learn this representation, but *also requires it to learn an additional distinct representation*. Specifically, we minimise the MI between these two representations so that the latter cannot rely on the same features. We extend this idea by adding representations, each time requiring the latest to be distinct from all previous representations.

**Downstream task: clustering**   For this work, our focus is on finding higher MI representations; we then assess the downstream capability on the ground truth task of image classification, meaning that we can either (1) learn a representation that must be 'decoded' via an additional learning step, or (2) produce a discrete labelling that requires no additional learning. Clustering methods offer a direct comparison and require *no labels for learning a mapping from the learned representation to class labels*. Instead, labels are only required to assign groups to appropriate classes and no learning is done using these. Our comparisons are with respect to clustering methods.

## 1.2   CONTRIBUTIONS

Learning a set of representations by encouraging them to have low MI, while still maximising the original augmentation-driven MI objective for each representation, is the core idea behind **D**eep **H**ierarchical **O**bject **G**rouping (DHOG). We define a mechanism to produce a set of hierarchically-ordered solutions (in the sense of easy-to-hard orderings, not tree structures). DHOG is able to better maximise the original MI objective between augmentations since each representation must correspond to a unique local optima. Our contributions are:

1. We demonstrate that current methods do not effectively maximise the MI objective[1] because greedy stochastic gradient descent (SGD) typically results in suboptimal local optima. To mitigate for this problem, we introducing DHOG: a robust neural network image grouping method to learn diverse and hierarchically arranged *sets of discrete image labellings* (Section 3) by explicitly modelling, accounting for, and avoiding spurious local optima, requiring only simple data augmentations, and needing no Sobel edge detection.

2. We show a marked improvement over the current state-of-the-art for standard benchmarks in end-to-end image clustering for CIFAR-10, CIFAR-100-20 (a 20-way class grouping of CIFAR-100, and SVHN; we set a new accuracy benchmarks on CINIC-10; and show the utility of our method on STL-10 (Section 4).

To be clear, DHOG still learns to map data augmentations to similar representations as this is imperative to the learning process. The difference is that DHOG enables a number of intentionally distinct data labellings to be learned, arranged hierarchically in terms of source feature complexity.

---

[1]We show this by finding higher mutual information solutions using DHOG, rather than by any analysis of the solutions themselves.

## 2 RELATED WORK

The idea of MI maximisation for representation learning is called the *infoMAX* principle (Linsker, 1988; Tschannen et al., 2019). Contrastive predictive coding (van den Oord et al., 2018) (CPC) models a 2D latent space using an autoregressive model and defines a predictive setup to maximise MI between distinct spatial locations. Deep InfoMAX (Hjelm et al., 2019) (DIM) does not maximise MI across a set of data augmentations, but instead uses mutual information neural estimation (Belghazi et al., 2018) and negative sampling to balance maximising MI between global representations and local representations. Augmented multiscale Deep InfoMAX (Bachman et al., 2019) (AMDIM) incorporates MI maximisation across data augmentations and multiscale comparisons.

Clustering approaches are more directly applicable for comparison with DHOG because they explicitly learn a discrete labelling. The authors of deep embedding for clustering (DEC) (Xie et al., 2016) focused their attention on jointly learning an embedding suited to clustering and a clustering itself. They argued that the notion of distance in the feature space is crucial to a clustering objective. Joint unsupervised learning of deep representations and image clusters (JULE) (Yang et al., 2016) provided supervisory signal for representation learning. Some methods (Ghasedi Dizaji et al., 2017; Fard et al., 2018) employ autoencoder architectures along with careful regularisation of cluster assignments to (1) ensure sufficient information retention, and (2) avoid cluster degeneracy (i.e., mapping all images to the same class).

Deep adaptive clustering (Chang et al., 2017) (DAC) recasts the clustering problem as binary pairwise classification, pre-selecting comparison samples via feature cosine distances. A constraint on the DAC system allows for a one-hot encoding that avoids cluster degeneracy. Another mechanism for dealing with degeneracy is to use a standard clustering algorithm, such as $K$-means to iteratively group on learned features. This approach is used by DeepCluster (Caron et al., 2018).

Associative deep clustering (ADC) (Haeusser et al., 2018) uses the idea that associations in the embedding space are useful for learning. A network was learned to associate data with (pseudo-labelled) centroids. They leveraged augmentations by encouraging samples to output similar cluster probabilities.

Deep comprehensive correlation mining (Wu et al., 2019) (DCCM) constructs a sample correlation graph for pseudo-labels and maximises the MI between augmentations, and the MI between local and global features for each augmentation. While many of the aforementioned methods estimate MI in some manner, invariant information clustering (Ji et al., 2019) (IIC) directly defines the MI using the $c$-way softmax output (i.e., probability of belong to class $c$), and maximises this over data augmentations to learn clusters. They effectively avoid degenerate solutions because MI maximisation implicitly targets marginal entropy. We use the same formulation for MI in Section 3.

## 3 METHOD

Figure 1 shows the DHOG architecture. DHOG is an approach for obtaining jointly trained multi-level representations as discrete labellings, arranged in a simple-to-complex hierarchy, and computed by separate 'heads'. A head is an unit that computes a multivariate class probability vector. By requiring low MI between heads, a diversity of solutions to the MI maximisation objective can be found. The head that best maximises MI between augmentations typically aligns better with a ground truth task that also relies on complex features that augmentations are designed to preserve.

Figure 1 demonstrates the DHOG architecture and training principles. There are shared model weights (②: ResNet blocks 1, 2, and 3) and head-specific weights (the MLP layers and ③: ResNet blocks 4 to 8). For the sake of brevity, we abuse notation and use $\mathrm{MI}(z, z')$ between labelling probability vectors as an overloaded shorthand for the mutual information $\mathrm{MI}(c, c')$ between the labelling random variables $c$ and $c'$ that have probability vectors $z$ and $z'$ respectively.

Any branch of the DHOG architecture (① to any $z_i$) can be regarded as a single neural network. These are trained to maximise the MI between the label variables at each head for different augmentations; i.e., between label variables with probability vectors $z_i(x)$ and $z_i(x')$ for augmentations $x$ and $x'$. Four augmentations are shown at ①. The MI is maximised pairwise between all pairs, at ④. This process can be considered *pulling* the mapped representations together.

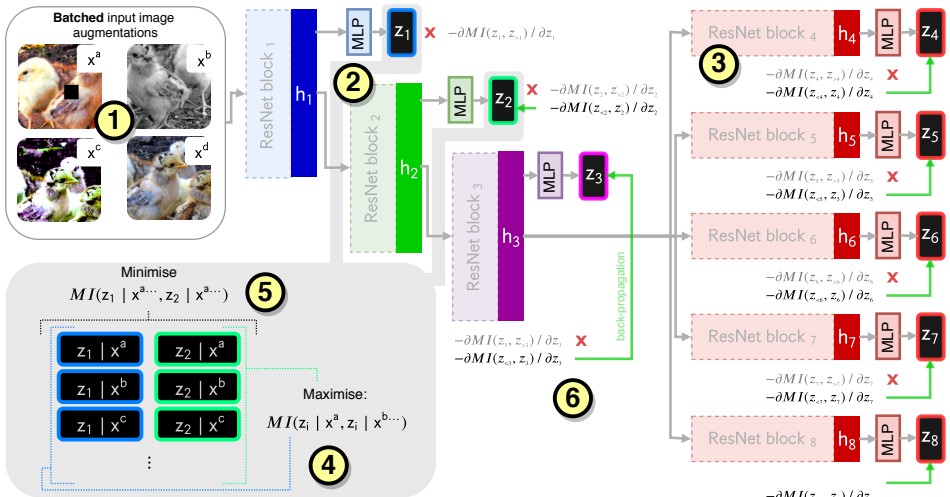

Figure 1: DHOG architecture. The skeleton is a ResNet18 (He et al., 2016). The final ResNet block is repeated $k - 3$ times ($k = 8$ here). ① Augmentations of each image, $x^{a \dots d}$, are separately processed by the network. ② Each shallow ResNet block $(1 \dots 3)$ constitutes shared computation for deeper blocks, while also computing separate probability vectors, $z_1 \dots z_3$. Each $z_i$ is viewed as the probability for each outcome of the random variable $c_i$ that makes a discrete labelling choice. ③ The deepest ResNet blocks compute further $z_{>3}$. ④ The network is trained by maximising the MI between allocations $c_i$ from *all data augmentations*, and ⑤ separately for each node $i$, minimising the MI between $c_i$ and $c_{<i}$ for the *same data augmentation*. ⑥ This is implemented by stopping gradients such that they are *not back-propagated* for later computation paths (red crosses).

Following IIC (Ji et al., 2019), we compute the MI directly from the label probability vectors within a minibatch. Let $\mathbf{z}_i, \mathbf{z}_i'$ denote the random probability vectors at head $i$ associated with sampling a data item and its augmentations, and passing those through the network. Then we can compute the mutual MI between labels associated with each augmentation using

$$\begin{aligned}
\mathrm{MI}_{aug}(c_i, c_i') = {} & \mathrm{Tr}(E[\mathbf{z}_i(\mathbf{z}_i')^T]^T \log(E[\mathbf{z}_i(\mathbf{z}_i')^T])) \\
& - E[\mathbf{z}_i^T] \log E[\mathbf{z}_i] - E[(\mathbf{z}_i')^T] \log E[(\mathbf{z}_i')],
\end{aligned} \tag{1}$$

where Tr is the matrix trace, logarithms are computed element-wise, and expectations are over data samples and augmentations of each sample. In practice we compute an empirical estimate of this MI based on samples from a minibatch.

## 3.1 DISTINCT HEADS

Each clustering head in DHOG is encouraged to compute unique solutions via *cross-head MI minimisation*. For a minibatch of images, the labelling from any head is optimised to have low MI with other heads' labellings. We assume multiple viable labellings because of natural patterns in the data. By encouraging low MI between heads, these must capture different patterns in the data.

Simple concepts (brightness, colour, etc.) are axes of variation that are reasonable and easy to group by. Groupings according to complex features (e.g., object type) typically require more processing and greedy optimisation may not discover these groupings without explicit encouragement. Unfortunately, the easier-to-compute groupings typically correspond poorly to downstream tasks. Without a mechanism to explore viable patterns in the data, greedy optimisation will avoid finding them.

**Cross-head MI minimisation** We address *suboptimal MI maximisation* by encouraging unique solutions at sequential heads ($z_1 \dots z_8$ in Figure 1), which rely on different features. Let $\mathbf{z}_i, \mathbf{z}_j$ denote the random probability vectors from two heads. We can *minimise* the MI across heads:

$$\begin{aligned}
\mathrm{MI}_{head}(c_i, c_j) = {} & \mathrm{Tr}(E[\mathbf{z}_i \mathbf{z}_j^T]^T \log(E[\mathbf{z}_i \mathbf{z}_j^T])) \\
& - E[\mathbf{z}_i^T] \log(E[\mathbf{z}_i]) - E[(\mathbf{z}_j)^T] \log(E[\mathbf{z}_j]).
\end{aligned} \tag{2}$$

Logarithms are element-wise, and expectations are over the data and augmentations. Note $\mathbf{z}_i$ and $\mathbf{z}_j$ are each computed from the *same data augmentation*. We estimate this from each minibatch sample. This process can be thought of as *pushing* the heads apart. We note that the $Tr$ operator is commutative – the hierarchical arrangement is accomplished through gradient stopping.

**Hierarchical arrangement**   Requiring $k$ heads (where $k = 8$ here) to produce unique representations is not necessarily the optimal method to account for suboptimal MI maximisation. Instead, what we do is encourage a simple-to-complex hierarchy structure to the heads, defined according to cross-head comparisons made using Equation 2. The hierarchy enables a reference mechanism to produce diverse labellings of the data.

Figure 1 shows 8 heads, 3 of which are computed from early residual blocks of the network. The hierarchical arrangement is induced by only updating head-specific weights according to comparisons made with earlier heads. In practice this is done by stopping the appropriate gradients – ⑥ and all red crosses. For example, when computing the MI between $\mathbf{z}_{i=6}$ and those using $\mathbf{z}_{i\neq 6}$, gradient back-propagation is allowed when $i < 6$ but not when $i > 6$. In other words, when learning to produce $\mathbf{z}_{i=6}$, the network is encouraged to produce a head that is distinct from heads 'lower' on the hierarchy. Extending this concept for $i = 1\dots 8$ gives rise to the hierarchy. Initial experiments showed that if this routine was ignored, the gains were reduced.

## 3.2   Objective

The part of the objective producing high MI representations by 'pulling' together discrete labellings from augmentations is Equation 1 normalised over $k$ heads:

$$\mathrm{MI}_{pull} = \frac{1}{k}\sum_{i=0}^{k} MI_{aug}(c_i, c_i'). \tag{3}$$

The quantity used to 'push' heads apart is Equation 2 normalised per head:

$$\mathrm{MI}_{push} = \sum_{i=1}^{k} \frac{\sum_{\substack{j=1\\j\neq i}}^{i} MI_{head}(c_i, c_j)}{i}, \tag{4}$$

where each cross-head MI term is scaled by the head index, $i$, since that directly tracks the number of comparisons made for each head. $i$ scales up the hierarchy, such that the total $\mathrm{MI}_{head}$ associated with any head is scaled according to *the number of comparisons*. Scaling ensures that head-specific weight updates are all equally important. The final optimisation objective is:

$$\theta^* = \arg\max_{\theta} \mathrm{MI}_{pull} - \alpha\mathrm{MI}_{push}, \tag{5}$$

where $\theta$ are the network parameters, $\alpha$ is a hyper-parameter we call the *cross-head MI-minimization coefficient*. For an ablation study we set $\alpha = 0$ in Section 4.

## 3.3   Design and training choices

The architecture (Figure 1) is based on a ResNet-18 backbone, where each residual block has two layers (with a skip connection over these). Blocks 1 to 3 have 64, 128, and 256 units, respectively. Each parallel final block (4 to 8, here) have 512 units. Each MLP has a single hidden layer of width 200. Early experiments showed that entire block repetition was important to model flexibility. Similar to IIC (Ji et al., 2019) we used four data augmentation repeats with a batch size of 220.

DHOG maximises MI between discrete labellings from different data augmentations. This is equivalent to a clustering and is similar to IIC. There are, however, key differences. In our experiments:

- We train for **1000 epochs** with a cosine annealing learning rate schedule.
- We **do not use sobel edge-detection** or any other preprocessing as a fixed processing step.
- We make use of the fast auto-augment CIFAR-10 data augmentation strategy (for all tested datasets) found by (Lim et al., 2019). We then randomly apply (with $p = 0.5$) grayscale and take random square crops of sizes 64 and 20 pixels for STL-10 and other datasets.

The choice of data augmentation is important, and we acknowledge that for a fair comparison to IIC the same augmentation strategy must be used. The ablation of any DHOG-specific loss (when $\alpha = 0$) largely recreates the IIC approach but with augmentations, network and head structure matched to DHOG; this enables a fair comparison between an IIC and DHOG approach.

Since STL-10 has much more unlabelled data of a similar but broader distribution than the training data, the idea of 'overclustering' was used by (Ji et al., 2019); they used more clusters than the number of classes (70 versus 10 in this case). We repeat each head with an overclustering head that does not play a part in the cross-head MI minimisation. The filter widths are doubled for STL-10. We interspersed the training data evenly and regularly through the minibatches.

To determine the DHOG cross-head MI-minimisation coefficient, $\alpha$, we carried out a non-exhaustive hyper parameter search using only CIFAR-10 images (without the labels), assessing performance on a held out validation set sampled from the training data. This did not use the evaluation data.

**Assessment**  Once learned, the optimal head can be identified either using the highest MI, or using a small set of labelled data. Alternatively all heads can be used as different potential alternatives, with posthoc selection of the best head according to some downstream task. In this paper the head that maximises the normalised mutual information on the training data is chosen. *This is then fully unsupervised*, as with the head selection protocol of IIC. We also give results for the best posthoc head to show the potential for downstream analysis.

## 4 EXPERIMENTS

The datasets used for assessment were CIFAR-10 (Krizhevsky et al., 2009), CIFAR-100-20 (CIFAR-100 (Krizhevsky et al., 2009) where classes are grouped into 20 super-classes), CINIC-10 (Darlow et al., 2018) (an extension of CIFAR-10 using images from Imagenet (Deng et al., 2009) of similar classes), street view house numbers (Netzer et al., 2011) (SVHN), and STL-10 (Coates et al., 2011). For CINIC-10 only the standard training set of 90000 images (without labels) was used for training.

Table 1 gives the accuracy, normalised mutual information (NMI), and the adjusted rand index (ARI) between remapped assignments and classification targets. Before assessment a labelling-to-class remapping was computed using the training data and the Hungarian method (Kuhn, 1955). The results listed for DHOG correspond to average over 3 seeded runs. At the time of writing, in terms of all measured metrics DHOG outperformed other relevant fully-unsupervised end-to-end clustering methods, with an accuracy improvement of $4.3\%$ on CIFAR-10, $0.35\%$ on CIFAR-100-20, and $7.2\%$ on SVHN. Importantly, no Sobel edge-detection was used.

We used an unsupervised posthoc head selection using $\text{NMI}(\mathsf{z}, \mathsf{z}')$ – *which corresponds directly to the original MI objective*. The selected heads almost always corresponded with the head that maxmimised $\text{NMI}(\mathsf{z}, \mathsf{y})$, where $\mathsf{y}$ are user-defined class labels. DHOG produces data groupings that:

1. **Better maximise the widely used MI objective** (between mappings of data augmentations) and therefore is an effective mechanism for dealing with suboptimal MI optimization owing to greedy SGD, as discussed in this work. Table 2 gives the NMI with and without DHOG (controlled by $\alpha$) to confirm this.

2. Correspond better with the challenging **underlying object classification test objective**.

It is only on STL-10 that DHOG never beat the current state-of-the-art. Our aim was to show that the simple hierarchical ordering of heads in DHOG improves performance. The difference between STL-10 with and without the MI cross-head minimisation term (controlled by $\alpha$) shows a marked improvement. IIC effectively used an overclustering procedure to achieve good results on STL-10.

The advantage of a hierarchical ordering is evident when considering the ablation study: with ($\alpha = 0.05$) and without ($\alpha = 0$) cross-head MI minimisation. Figure 2 (a) and (b) are accuracy versus head curves, showing that without cross-head MI minimisation later heads converge to similar solutions. The confusion matrices in Figure 3 (b) show the classes the final learned network confuses in CIFAR-10. Compare this to the confusion matrix in Figure 3 (a) where $\alpha = 0$ and note the greater prevalence of cross-class confusion.

| | Method | Accuracy | NMI(z, y) | ARI |
|---|---|---|---|---|
| CIFAR-10 | $K$-means on pixels | $21.18 \pm 0.0170$ | $0.0811 \pm 0.0001$ | $0.0412 \pm 0.0001$ |
| | Cartesian K-means | $22.89$ | $0.0871$ | $0.0487$ |
| | JULE | $27.15$ | $0.1923$ | $0.1377$ |
| | DEC [†] | $30.1$ | - | - |
| | DAC | $52.18$ | $0.3956$ | $0.3059$ |
| | DeepCluster [†] | $37.40$ | - | - |
| | ADC | $29.3 \pm 1.5$ | - | - |
| | DCCM | $62.3$ | $0.496$ | $0.408$ |
| | IIC [†] | $61.7$ | - | - |
| | DHOG ($\alpha = 0$, ablation) | $57.49 \pm 0.8929$ | $0.5022 \pm 0.0054$ | $0.4010 \pm 0.0091$ |
| | DHOG, unsup. ($\alpha = 0.05$) | $\mathbf{66.61 \pm 1.699}$ | $\mathbf{0.5854 \pm 0.0080}$ | $\mathbf{0.4916 \pm 0.0160}$ |
| | DHOG, best ($\alpha = 0.05$) | $66.61 \pm 1.699$ | $0.5854 \pm 0.0080$ | $0.4916 \pm 0.0160$ |
| CIFAR-100-20 | $K$-means on pixels | $13.78 \pm 0.2557$ | $0.0862 \pm 0.0012$ | $0.0274 \pm 0.0005$ |
| | DAC [†] | $23.8$ | - | - |
| | DeepCluster [†*] | $18.9$ | - | - |
| | ADC | $16.0$ | - | - |
| | IIC [†] | $25.7$ | - | - |
| | DHOG ($\alpha = 0$, ablation) | $20.22 \pm 0.2584$ | $0.1880 \pm 0.0019$ | $0.0846 \pm 0.0026$ |
| | DHOG, unsup. ($\alpha = 0.05$) | $\mathbf{26.05 \pm 0.3519}$ | $\mathbf{0.2579 \pm 0.0086}$ | $\mathbf{0.1177 \pm 0.0063}$ |
| | DHOG, best ($\alpha = 0.05$) | $27.57 \pm 1.069$ | $0.2687 \pm 0.0061$ | $0.1224 \pm 0.0091$ |
| CINIC-10 | $K$-means on pixels | $20.80 \pm 0.8550$ | $0.0378 \pm 0.0001$ | $0.0205 \pm 0.0007$ |
| | DHOG ($\alpha = 0$, ablation) | $\mathbf{41.66 \pm 0.8273}$ | $0.3276 \pm 0.0084$ | $\mathbf{0.2108 \pm 0.0034}$ |
| | DHOG, unsup. ($\alpha = 0.05$) | $37.65 \pm 2.7373$ | $\mathbf{0.3317 \pm 0.0096}$ | $0.1993 \pm 0.0030$ |
| | DHOG, best ($\alpha = 0.05$) | $43.06 \pm 2.1105$ | $0.3725 \pm 0.0075$ | $0.2396 \pm 0.0087$ |
| SVHN | $K$-means on pixels | $11.35 \pm 0.2347$ | $0.0054 \pm 0.0004$ | $0.0007 \pm 0.0004$ |
| | ADC | $38.6 \pm 4.1$ | - | - |
| | DHOG ($\alpha = 0$, ablation) | $14.27 \pm 2.8784$ | $0.0298 \pm 0.0321$ | $0.0209 \pm 0.0237$ |
| | DHOG, unsup. ($\alpha = 0.05$) | $\mathbf{45.81 \pm 8.5427}$ | $\mathbf{0.4859 \pm 0.1229}$ | $\mathbf{0.3686 \pm 0.1296}$ |
| | DHOG, best ($\alpha = 0.05$) | $49.05 \pm 8.2717$ | $0.4658 \pm 0.0556$ | $0.3848 \pm 0.0557$ |
| STL-10 | $K$-means on pixels | $21.58 \pm 0.2151$ | $0.0936 \pm 0.0005$ | $0.0487 \pm 0.0009$ |
| | JULE [†] | $27.7$ | - | - |
| | DEC | $35.90$ | - | - |
| | DAC | $46.99$ | $0.3656$ | $0.2565$ |
| | DeepCluster [†] | $33.40$ | - | - |
| | ADC | $47.8 \pm 2.7$ | - | - |
| | DCCM | $48.2$ | $0.376$ | $0.262$ |
| | IIC [†] | $\mathbf{59.6}$ | - | - |
| | DHOG ($\alpha = 0$, ablation) | $38.70 \pm 4.4696$ | $0.3878 \pm 0.0331$ | $0.2412 \pm 0.0265$ |
| | DHOG, unsup. ($\alpha = 0.05$) | $48.27 \pm 1.915$ | $\mathbf{0.4127 \pm 0.0171}$ | $\mathbf{0.2723 \pm 0.0119}$ |
| | DHOG, best ($\alpha = 0.05$) | $48.27 \pm 1.915$ | $0.4127 \pm 0.0171$ | $0.2723 \pm 0.0119$ |

Table 1: Test set results on all datasets, taken from papers where possible. Results with [†] are from (Ji et al., 2019). $\mathrm{NMI}(z, y)$ is between remapped predicted label assignments and class targets, $y$. Ablation: DHOG with $\alpha = 0.0$ **is the IIC method but with conditions/augmentations matching those of DHOG($\alpha = 0.5$)**. Our method is DHOG with $\alpha = 0.05$, highlighted in blue. We give results for the head chosen for the best $\mathrm{NMI}(z, z')$ and the head chosen for the best $\mathrm{NMI}(z, y)$. In most cases max MI chooses the optimal head.

| Dataset | NMI(z, z'), $\alpha = 0$ | NMI(z, z'), $\alpha = 0.05$ |
|---|---|---|
| CIFAR-10 | $0.7292 \pm 0.0066$ | $0.7994 \pm 0.0081$ |
| CIFAR-100-20 | $0.6104 \pm 0.0098$ | $0.6506 \pm 0.0040$ |
| CINIC-10 | $0.6408 \pm 0.0015$ | $0.6991 \pm 0.0044$ |
| SVHN | $0.6337 \pm 0.0085$ | $0.7265 \pm 0.0093$ |
| STL-10 | $0.6713 \pm 0.0175$ | $0.6610 \pm 0.0084$ |

Table 2: NMI between representations of augmentations for the training dataset, evidencing that cross-head MI minimisation does indeed aid in avoiding local optima of this objective function.

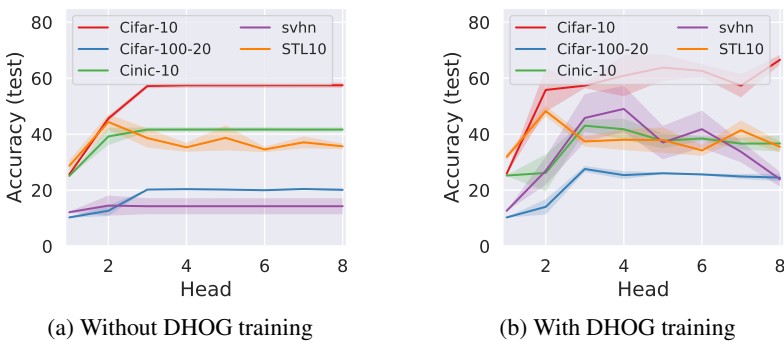

(a) Without DHOG training      (b) With DHOG training

Figure 2: Accuracy per head. DHOG causes heads to learn distinct solutions.

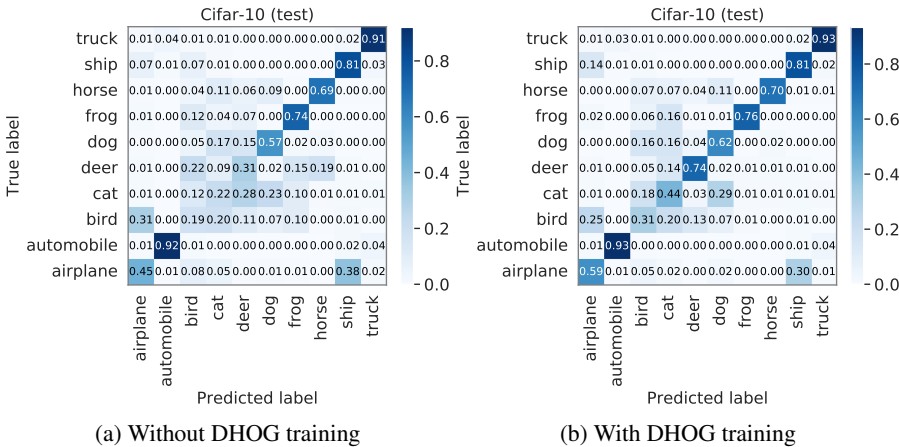

(a) Without DHOG training      (b) With DHOG training

Figure 3: Confusion matrices from the same seed (a) without and (b) with DHOG cross-head MI minimisation. These networks struggle with distinguishing natural objects (birds, cats, deer, etc.), although DHOG does improve this.

A claim throughout this paper is that greedy training of neural networks can result in sub-optimal MI maximisation. Table 2 shows that for all datasets except STL-10 (for which further experiments are needed) DHOG resulted in a better MI result, thereby directly improving the training objective.

## 5 CONCLUSION

We presented deep hierarchical object grouping (DHOG): a method that leverages the challenges faced by current data augmentation-driven unsupervised representation learning methods that maximise mutual information. Learning a good representation of an image using data augmentations is limited by the user, who chooses the set of plausible data augmentations but who is also unable to cost-effectively define an ideal set of augmentations. We argue and show that learning using greedy optimisation typically causes models to get stuck in local optima, since the data augmentations fail to fully describe the sought after invariances to all task-irrelevant information.

We address this pitfall via a simple-to-complex ordered sequence of representations. DHOG works by minimising mutual information between these representations such that those later in the hierarchy are encouraged to produce unique and independent discrete labellings of the data (w.r.t. earlier representations). Therefore, later heads avoid becoming stuck in the same local optima of the original mutual information objective (between augmentations, applied separately to each head). Our tests showed that DHOG resulted in an improvement on CIFAR-10, CIFAR-100-20, and SVHN, without using preprocessing such as Sobel edge detection, and a consistent improvement of the underlying MI objective.

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
