# OpenReview forum: "DHOG: Deep Hierarchical Object Grouping"
_ICLR.cc/2021/Conference — Reject_

### Official Review · AnonReviewer4 · 2020-10-19
**Novel paper, nice results**

**Rating:** 4
**Confidence:** 2

**Review:**

In this paper, the authors study the problem of unsupervised representation learning from data augmentations. Specifically, the authors claim that existing methods are prone to getting stuck at local minima owing to easy-to-learn local representations that optimise the commonly used MI objectives, and then propose a hierarchical method that tackles optimisation at multiple layers of the feature hierarchy.

Strengths:
- A highly novel solution to an important problem in representation learning.
- Strong results obtained with respect to the state of the art.
- Extensive analysis and comparisons.

Weaknesses:
- "We demonstrate that current methods do not effectively maximise the MI objective" + footnote: "We show this by finding higher mutual information solutions using DHOG, rather than by any analysis of the solutions themselves." => This claim calls for an analysis of the solutions and since this is missing in the paper, I would rephrase the claim.

- It would have been nicer to have experimental analysis on different features (e.g. colour) being more prone to local optima, though this is intuitive. This could have significantly increased the impact of the paper.

Minor comments:
- "a reasonable mapping need only compute colour information" => "a reasonable mapping needs only compute colour information".
- "Learning a set of representations by encouraging them to have low MI," => This should be high MI?
- "CIFAR-10, CIFAR-100-20 (a 20-way" => The parenthesis is not closed.
- "A network was learned to associate" => "A network was trained to associate".

**AFTER AUTHOR RESPONSE**

I have read the other reviewers carefully and the feedback provided by the authors. The reviewers had two major concerns: (i) Theoretical or empirical justification/proof for the following claim (and the motivation) of the paper: “the current methods do not effectively maximize the MI objective because greedy SGD typically results in suboptimal local optima”. (ii) Lack of comparisons with newer methods from e.g. ECCV2020 etc.

For the second, I feel empathy with the authors: In such a rapidly progressing field, it is difficult to integrate comparisons with new methods that are published while writing/submitting a paper. I am sure all of us have experienced similar problems.

For the first issue, I disagree with the authors and agree with the other reviewers: This is an important claim that needs to be justified and I disagree with the authors's comment on the current results of the paper being a sufficient empirical evidence for the claim. An ICLR paper should have provided the necessary evidences and justifications.

---

> ### Author Response · Authors · 2020-11-18
> **Response to AnonReviewer4**
>
>
> (1)
> We wished to state that in order to claim that current methods do not effectively maximise the MI objective it is sufficient to show that, in general, an alternative, better optima of that objective can be found by an alternative method (in this case by DHOG). The analysis is effectively the assessment of MI (via NMI in the results table). We will rephrase the footnote to better express this in light of your comment.
>
> (2)
> It is hard to know how to directly assess whether a particular form of feature is more or less likely to be a local optima solution, partly as it is hard to characterise the feature space in human-understandable terms. You are right that we could look at colour, but at the moment we do not know how to generalise that sufficiently well to more complex features to make a robust analysis of this form. We agree it would be excellent if there were a way to do this.
>
>
> Minor comments
> (1) Thank you for checking our paper for low-level issues - we do appreciate it. In the case of "a reasonable mapping need only compute colour information" our usage here is correct - this uses a non-inflective auxiliary verb - a helper to the "compute". See e.g., https://english.stackexchange.com/questions/297233/is-one-needs-only-or-one-need-only-correct/297235 for one particular explanation of this usage.
> (2) No, we want the MI between the set of representations to have low MI, while still having high MI across augmentations.
> (3) Fixed.
> (4) Fixed.

---

### Official Review · AnonReviewer2 · 2020-10-23
**An extension to Invariant Information Clustering (IIC) aiming to provide more informative solutions**

**Rating:** 6
**Confidence:** 4

**Review:**

The paper presents an extension to the Invariant Information Clustering (IIC) methodology for image clustering.
IIC is an end-to-end deep clustering methodology that directly provides the cluster label of an input image.
Training is based on maximizing Mutual Information (MI) between pairs of related images (typically generated through augmentations).
In this work the authors argue that IIC performs suboptimal maximization of MI, thus the obtained solutions focus mainly on low-level representations and do not distill higher level semantic similarity of images.
To solve this problem, they propose an IIC extension (called DHOG) where multipe clustering solutions are obtained through MI maximization, under the constraint that the solutions found are sufficiently different (ie. minimum MI between the clustering solutions is seeked).

The proposed objective function is novel, however it involves a regularization constant \alpha to be set by the user.

The idea of stopping gradients to impose a hierarchy of solutions is interesting.

The experimental results indicate improved clustering performance.

Comments to be addressed:
1) Application of the method requires the specification of the number of K heads. It in this work K=8 is set. What is the reasoning for such selection and how does the value of K affect the results?
2) DHOG is computatonally more complex compared to IIC. What is the execution time of DHOG compared to IIC?
3) What happens for values of \alpha greater that 0.05? How sensitive is the method on the value of \alpha?
4) It is strongly suggested to include a visual example illustrating the difference among the obtained clustering solutions (e.g. on CIFAR-10) to provide visual evidence that improved clustering solutions are progressively found.
5) It is not clear whether IIC results are obtained using Sobel preprocssing or not. What if Sobel preprocessing is used in DHOG?
6) The presentation of the paper could be improved, by first briefly describing the IIC approach and then presenting the proposed extension (DHOG).

---

> ### Author Response · Authors · 2020-11-18
> **Response to AnonReviewer2**
>
> (1)
> K was set using a small number of cross-validation runs (similar to how we set alpha). The reasoning is simply a question of balancing the driving motivation of this research (diverse representation learning) and the compute capacity we had at our disposal. Since adding heads incurs memory and compute costs, adding more heads (after 8) yielded diminishing returns - the solutions became more similar to earlier solutions in the hierarchy. Because of this, the results were not affected when K > 8: we simply used the highest number of heads that we could.
>
> (2)
> The backbone network of DHOG is a ResNet18. The backbone of IIC is a ResNet34. While the additional heads mean that the parameter counts are similar between IIC and DHOG, we can discard K-1 heads after learning - only the single head that best optimises the training objective is needed for inference. The execution times should be very similar, although we have not tested this directly.
>
> (3)
> We explored alpha after setting it using cross-validation on Cifar-10 since it was not sensitive within an order of magnitude. As long as the DHOG method contributed to learning representations (alpha > 0, within reason), the results were largely similar.
>
> (4)
> We agree that this is indeed informative. We originally prepared this and wanted to include it in the main paper but were thwarted by space constraints. We have now added it as an appendix, and apoligise for its original omission.
>
> (5)
> None of the results presented in our work used Sobel preprocessing. Our argument for not using Sobel is that it is destructive and can easily remove useful information (such as colour), even though the gains it offers are clear for other methods. Further, edge detection is only meaningful for natural images or related - in other settings (e.g. some medical imaging settings) edge detection is not a sensible preprocessor. The nature of this work is such that we do not want to cloud the gains owing to DHOG with heuristic, symptomatic treatment of the underlying issue. We do outline this in Section 1.1 under "Dealing with greedy solutions".
>
>
> (6)
> We expect that we have mistakenly given the impression that our work can be contextualised primarily as an extension to IIC, and we understand why. However, the fundamental idea behind creating diverse representations can be applied to any MI-based unsupervised learning method. As other reviewers have pointed out, there are some modern methods that outperform IIC and DHOG (and we have included these in the paper now, while also making clear the context of our work). Our focus, however, is to show the efficacy of learning diverse representations, and not necessarily beating state of the art. We are planning to apply DHOG to these methods, but we would also like to make clear that our main goal is to introduce a method that learns diverse representations, and in so doing helps to better optimise the MI objective. This is a key problem of neural networks for unsupervised representation learning, and DHOG is an approach to enable better representation learning. Built on a superior backbone (like SCAN, for example), we might hope that DHOG will nonetheless improve results on many of the methods that are now being proposed. In our opinion the long term benefits or otherwise will be easier to assess through the publication of this approach and the discussion and analysis that will come from that.

---

### Official Review · AnonReviewer1 · 2020-10-28
**Review #1**

**Rating:** 3
**Confidence:** 4

**Review:**

<Summary>

This paper addresses the problem of unsupervised learning of class representation using data augmentation. Its key idea is to encourage the learned representations to have low MI while maximizing the original augmentation-driven MI objective. It reports the improved performance for the benchmarks of Ji et al. 2019 – classification on some easy datasets (e.g. CIFAR-10, CINIC-10, SVHN and STL-10).

<Strengths>

1. The proposed approach proposes an MI minimization strategy for unsupervised clustering that can obtain more informative representation than previous methods.

2. The proposed DHOG method shows better performance than some baselines reported in  Ji et al. 2019 for some easy classification tasks on  CIFAR-10, CINIC-10, SVHN and STL-10.

<Weakness>

1. This work summarizes the two contributions in sec 1.2. However, both of them are not fully supported.

(1) This work claims that “the current methods do not effectively maximize the MI objective because greedy SGD typically results in suboptimal local optima.” However, there is neither theoretical justification nor empirical results that show why the proposed approach does not suffer from this issue and why it is subsequently better than other works.

(2) This argument may be based on that the proposed DHOG adopts distinct head that encourages different solutions between heads one another and a simple-to-complex hierarchy structure to the heads. Conceptually, they could be reasonable solutions to the issue, but there is no theoretic and empirical evidence that supports this argument.

(3) The other contribution that this work claims is strong performance of unsupervised clustering on benchmark datasets. However, the empirical comparison is only carried out with baselines in Ji et al., 2019.

(4) More critically, this work fails to cite the following recently published papers that reported much stronger performance on the same datasets.

a. W. Van Gansbek et al, SCAN: Learning to Classify Images without Labels, ECCV 2020.
b. G. Shiran & D. Weinshall  Multi-Modal Deep Clustering: Unsupervised Partitioning of Image, ICLR 2021.
c. N. Astorga et al. MPCC: Matching Priors and Conditionals for Clustering, ECCV 2020.

<Conclusion>

My initial decision toward this submission is rejection because the main contributions of this work are not fully convinced. Moreover, this work ignores key related papers that show better performance on the same benchmarks.

---

> ### Author Response · Authors · 2020-11-18
> **Response to AnonReviewer1**
>
> (1)
> Theoretical justification for many aspects of neural networks are considered a significant and unsatisfied challenge and we suspect that consideration of that would be a separate paper in its own right. Hence we focus on (a) empirical evidence and (b) give evidence for a description of what is going on. Our goal is to show that many current methods do not pay heed to the tendency of neural networks to end up in poor local optima, particularly for unsupervised MI-based learning. We demonstrate they do not sufficiently optimise the MI objective by providing an alternative that finds a much better optimum for the training objective - it is not just that it generalises better. In order to demonstrate the suboptimality of an optimizer, it is sufficient to show that there is another optimizer that performs better. We do that and demonstrate it using NMI in Table 1. Though we show that other approaches do suffer from this issue, we do not claim that our approach perfectly fixes this - rather we just demonstrate empirically that it handles the problem significantly better.
>
> (2)
> As you say, we propose a particular approach that was designed precisely to mitigate the particular issue that we identified with current methods. Though we prove no theorems about this (that is not the target of the paper), we do demonstrate the difference in the form of representations when we do or do not use a DHOG approach. The results in Table 1 provide clear empirical evidence supports an improvement in representation (both in terms of MI and in terms of downstream tasks). The ablations show that DHOG is critical to this issue. We would be very happy to provide specific metrics the reviewer thinks would be useful beyond those already supplied.
>
> Even if the reviewer disagrees that our argument is the reason for the improved performance, we think the identification of poor-overly simple representations being learnt by many current MI optimizers is an important topic for onward discussion at the conference, and that this paper is an important contribution to that discussion.
>
>
> (3) and (4)
> Thank you for your comments and your pointers. We agree that there are recent papers that we should now adapt the method for and compare with. The approach of enforcing independence of a hierarchy of heads is complementary to any other choice of method: it is an operator to apply to other methods, not a single choice of architecture. At the time of creating DHOG, and consequently writing this paper, the state-of-the-art neural network clustering method was IIC (Ji et al., 2019). Sadly SCAN or MPCC were not available and accessible when we were working on this. However we accept the need to keep up with the recent changes, and we have adapted the paper to fully acknowledge these papers and to give comparisons. We do emphasise though, that the primary objective of this paper is not to chase state of the art. What we are showing with this work is that by developing methods that specifically target learning a diverse set of representations we can better optimise a MI objective. We applied this to IIC at the time, and that is the context within which we present this work, and we look forward to assessing its continued application of DHOG to the very recent methods as part of our future work.
>
> Almost separate from this particular paper, we also think that it is important that we all, as proponents in a fast-moving field, recognise that researchers' efforts, findings, and insights should not be nullified in the context that others, working in parallel, also had different good ideas which happened to do as well or better on benchmarks (though benchmarks are important). In our opinion that would be a sad state of affairs for this research community and have a narrowing effect on the field which we feel would hinder it in the future.
>
> In light of the helpful comments, observations and requests for clarification, we have modified the claims of the paper and made the above-mentioned context clear. We have also included additional comparisons. We hope that alleviates your concerns and sheds an important perspective on the paper.

---

### Official Review · AnonReviewer3 · 2020-10-29
**MI based unsupervised representation learning**

**Rating:** 4
**Confidence:** 4

**Review:**

Authors introduce a method which is tailored to unsupervised representation learning via minimizing mutual information. Authors support their method with an experimental study. The proposed method has some merits like finding a representation that has higher mutual information compared to baselines.
I have several concerns about the paper:
a)	Authors state that some greedy optimization may end up lower MI optima. Since deep neural networks are not convex, is this a surprising point or is this expected one?
b)	I think empirical comparison needs significant improvement. Authors mention the proposed method was outperforming the state of the art “at the time of writing”. However, currently I believe this is not the case. For example, GATCluster achieves 28.1% on Cifar100-20.
c)	I see the proposed method as shown in Fig 1, is novel. As far as I understand it is adding some extra blocks to already existing method and calculates mutual information between heads. I would like to point out that hierarchical ordering is an interesting idea however the impact of hierarchical ordering is not very clear. Why does architecture need extra heads i.e. h_4 to h_8.  Would you please emphasize the novelty of the idea bit more?
Although I have some concerns about the paper, I would like to be extremely clear that I am open to change my view if more explanation and/or evidence supplied.

---

> ### Author Response · Authors · 2020-11-18
> **Response to AnonReviewer3**
>
> a) You are absolutely right that it is expected that neural networks will end up in a local optima. However the quality of the local optima matters (how suboptimal it is). We show that the local optima that standard neural networks with SGD find are substantially poorer local optima than an approach (such as DHOG) which counteracts the overly-simple solutions they produce. This is true for the actual target training objective, not just the generalisation capability, though that is impacted too.
>
> b) We understand that there are now several methods that warrant comparison in this context. This approach was developed while IIC was the top-performing clustering method and structured our comparisons in that context. So strictly it was state-of-art at the time of writing, but you are right that we should temper that claim. Though time moves on quickly in this field, the approach of enforcing independence of a hierarchy of heads is complementary to any other choice of method: it is an operator to apply to other methods, not a single choice of architecture - the improvements over IIC indicate the benefit to be obtained in one setting.
>
> In addition, the fundamental idea of diverse solutions being key to better MI objective optimisation (see the point below) stands as an important research finding, regardless of the underlying clustering method employed. We urge you to consider the context of our work, and that what we were trying to achieve is a method of learning diverse representations - chasing the state-of-the-art is not our primary driver and we hope that this does not obscure the utility of what we have achieved.
>
> We have modified the claim in the paper to make it clear there are now other methods available and tempered the claim that this was state-of-the art, and will provide these additional comparisons. Nonetheless, in the future we will adapt DHOG to the current bleeding edge of deep clustering methods to explore how diverse representation learning can improve these.
>
> c) We mentioned toward the end of Section 3.1 that without the hierarchy the gains were reduced. A good way of understanding the hierarchy is to assume that the earliest head (h_1) is free to learn any representation that satisfies the MI objective. Since the basis of our argument is that this is suboptimal (and indeed it must be a simple-to-compute representation as it is early in the network), this head becomes a point of comparison to enable learning. That is, the next head (h_2) must have low MI with h_1. In so doing h_2 avoids the same suboptimal local optima that h_1 ends up in. We simply assume that there are a number of viable poorer local optima, and extend this line of reasoning with additional heads (the number is largely arbitrary) to help explore potentially better ones. The hierarchical approach to learn different representations pushes each head to be conditionally independent of all previous heads (the gradient stopping ensures that later heads don't influence earlier ones directly), and therefore diverse. The hierarchy was critical to this process and the results achieved. Learning diverse representations is a key problem in the use of neural networks for unsupervised representation learning.

---

### Author Response · Authors · 2020-11-20
**State-of-the-art comparisons**

As reviewers pointed out there are now additional comparison works that have recently been made available and were not included in our original submission. This is because we were working on DHOG, and consequently writing this paper, concurrently with that research. Using deep neural networks for clustering has recently become a popular research topic, and while it is difficult to keep up with advances while nonetheless exploring good, interesting ideas, we acknowledge that we could have been more clear regarding the context and application of DHOG.

Instead of being a new clustering method, DHOG is an operator that enables learning of diverse representations in the MI-based learning paradigm, and particularly clustering. When we were developing our approach IIC was the state-of-the-art and benefits over IIC demonstrate the utility of DHOG. It is sensible to apply DHOG to even more modern clustering approaches (like SCAN, MPCC, GATCluster, etc.), but these were not readily available while we were working on DHOG.

The reliance on easy-to-compute features in the data is a key problem in the area of unsupervised representation learning, and DHOG is our contribution to tackling that problem in a systematic fashion. We hope that the state-of-the-art achievements presented in recent work in the neural network clustering domain do not overshadow our findings and insights, as this would have a narrowing effect on the broader field and hinder it in the future.

---

### Decision · Program_Chairs · 2021-01-07
**Final Decision**

**Decision:**

Reject

**Comment:**

During the discussion phase, although the reviewers acknowledge superior empirical performance of the proposed method, they shared the two major concerns:
1. Lack of theoretical or empirical justification/proof for the key statement: "the current methods do not effectively maximize the MI objective because greedy SGD typically results in suboptimal local optima".
1. Lack of comparisons with newer methods from e.g. ECCV2020 etc.

In particular, the first point is crucial. As the reviewers pointed out, since it is the main contribution and the key message of this paper, it should be carefully examined theoretically and/or empirically. However, in its current state, there is no theoretical analysis, and empirical evaluation is not convincing.

About the second point, although I think it cannot be a solo reason for rejection, at least it is better to cite and discuss it fo the completeness.

Overall, the contribution of this paper it not significant enough for publication. Hence I will reject the paper.